# Distance-dependent seed–seedling transition in the tree *Castanopsis sclerophylla* is altered by fragment size

Rong Wang [1], Yi-Su Shi[1], Yu-Xuan Zhang[1], Gao-Fu Xu[2], Guo-Chun Shen[1,3] & Xiao-Yong Chen[1,3]

Negative distance-dependence of conspecific seedling mortality (NDisDM) is a crucial stabilizing force that regulates plant diversity, but it remains unclear whether and how fragment size shifts the strength of NDisDM. Here, we surveyed the seed–seedling transition process for a total of 25,500 seeds of a local dominant tree species on islands of various sizes in a reservoir and on the nearby mainland. We found significant NDisDM on the mainland and large and medium islands, with significantly stronger NDisDM on medium islands. However, positive distance-dependent mortality was detected on small islands. Changes in distance-dependence were critically driven by both rodent attack and pathogen infestation, which were significantly affected by fragment size. Our results emphasize the necessity of incorporating the effects of fragment size on distance-dependent regeneration of dominant plant species into the existing frameworks for better predicting the consequences of habitat fragmentation.

[1] Zhejiang Tiantong Forest Ecosystem National Observation and Research Station, School of Ecological and Environmental Sciences, East China Normal University, 200241 Shanghai, China. [2] Xin'an River Development Corporation, 311700 Chun'an, China. [3] Shanghai Institute of Pollution Control and Ecological Security, 200092 Shanghai, China. Correspondence and requests for materials should be addressed to G.-C.S. (email: gcshen@des.ecnu.edu.cn) or to X.-Y.C. (email: xychen@des.ecnu.edu.cn)

The Janzen–Connell hypothesis proposes that offspring mortality of a plant species will increase with decreased distance to its conspecific adults because its natural enemies tend to occur at high density in the vicinity of the adults, leading to a commonly expected negative distance-dependent pattern of mortality in the transition from seed to seedling[1,2]. This negative distance-dependence of mortality (NDisDM) is recognized as an important stabilizing force to regulate the seedling abundance of dominant species[3–5] and has been found in many types of natural communities[2,6,7]. However, the strength of NDisDM varies widely among and within species, and the ecological drivers of such variation remain unclear[6,8–10].

Fragment size contributes critically to shaping biotic conditions[11–13] and thus may profoundly affect the strength of NDisDM, but its potential role has been largely ignored in previous studies. Variations in fragment size can introduce strong fluctuations in natural enemies (e.g., seed predators and herbivores) of plants, as a consequence of the relative importance of regulations (i.e., top-down and bottom-up controls) stabilizing food web interactions[14–16]. For example, absence of top predators in small habitat fragments can stimulate a rapid increase in density of consumers such as herbivores and seed predators[17], likely leading to either strengthened or weakened NDisDM in seed and seedling stages of plants, depending on whether the densities of specific natural enemies of plants mainly increase nearby their host trees. In addition, NDisDM strength may decrease in small forest fragments because low species richness and reduced population sizes of plants probably fail to support specialized consumers[11,18–21], but an inverse pattern is likely to occur when low plant diversity facilitates host-specific natural enemies if their host plants become dominant[22].

Varying fragment size can simultaneously alter abiotic environments[13,15], and thereby may also influence the strength of NDisDM[8,10]. Generally, edge effects are strengthened with decreasing fragment size, turning most areas in small habitat fragments into edge zones, where seeds and seedlings of plants adapting to the interior environments in forests may suffer dramatically high mortality due to the deteriorated environment[15,23]. Such high seed/seedling mortality mirrors the failed regeneration of these plants when their propagules are dispersed over long distances, weakening the strength of NDisDM and probably even resulting in a shift from NDisDM to positive distance-dependence (PDisDM) of mortality. Additionally, variations in abiotic conditions can influence the density and spatial distribution of herbivores and seed predators in small forest patches[24,25], and thereby generate a complex effect on NDisDM strength.

Despite these potential effects of fragment size on NDisDM and the large variation of NDisDM strength among studies, few studies have tested the relationship between fragment size and NDisDM strength.

Here, we focused on a dominant tree species, *Castanopsis sclerophylla* (Fagaceae), on islands of varying sizes. We found that the relative abundance of *C. sclerophylla* was high and did not vary significantly among the different fragment types and that local rodents were the major seed dispersers and seed predators of Fagaceae, but with a strong feeding preference for *C. sclerophylla* seeds. We tracked *C. sclerophylla* seeds through the seed–seedling transition process for four consecutive years. We detected significant NDisDM on the mainland and large and medium islands at all seed–seedling transition stages with significantly stronger NDisDM on medium islands, but PDisDM at seed-dispersal and seedling-emergence stages on small islands. We conclude that fragment size is an important factor determining the direction and strength of distance-dependent regeneration of *C. sclerophylla*. Our results suggest the role of fragment size in shaping the spatial distribution of other local dominant plants.

## Results

**Impacts of fragment size on distance-dependence mortality.** We identified islands on which *C. sclerophylla* are the dominant tree species and categorized them into large (875 ha), medium (13–51 ha) and small (1.1–3.9 ha) islands within a reservoir, the Thousand-Island Lake (TIL), and the surrounding mainland (>300 ha) (Fig. 1a). The relative abundance of *C. sclerophylla* was high and did not vary significantly among the different fragment types. Local rodents were the major seed dispersers and predominant seed predators of *Fagaceae* seeds but showed a strong feeding preference for *C. sclerophylla* seeds (see Methods and Supplementary Table 1). For four consecutive years, we tracked *C. sclerophylla* seeds through the seed–seedling transition process, comprising seed-dispersal, over-winter and seedling-emergence stages (see Methods and Fig. 1b) to determine: any evidence of NDisDM at early life stages in our study system; how direction and strength of distance-dependence changed with fragment size; and how the strength of distance-dependent mortality of seed/seedlings responded to rodent density.

We successfully tracked 10,261 out of 25,500 seeds of *C. sclerophylla* used in the four-year seed–seedling transition experiments. The remaining seeds were mostly hoarded in underground rat holes and were therefore unlikely to contribute to seed–seedling transition (see Methods). Only 3.1% of the tracked seeds survived and germinated at the end of our experiments each year (Supplementary Fig. 1a).

We tested whether strength of distance-dependent mortality — represented by the regression slope between seed attack rate or seed/seedling mortality rate and the distance from a seed to the nearest conspecific tree (DTNCT) (see Fig. 1c and Methods) — at each seed–seedling transition stage varied among different fragment types using generalized linear mixed models (GLMMs)[26]. We found significant negative relationships ($p < 0.001$ in all tests; Fig. 2a–c; Table 1) between seed attack/seedling mortality rate and DTNCT, i.e. showing NDisDM, at all seed–seedling transition stages on the mainland and the large island, with similar strengths in these two fragment types ($p > 0.05$ in all tests; Fig. 2d; Table 1). Relative to the larger fragments, NDisDM was significantly strengthened on the medium islands at all seed–seedling transition stages ($p < 0.05$ in all tests; Fig. 2d; Table 1). Contrastingly, significant PDisDM was observed at seed-dispersal and seedling-emergence stages on the small islands ($p < 0.05$ in both tests; Fig. 2a, c; Table 1); and, compared with other fragment types, there was a far weakened but significant NDisDM pattern at the over-winter stage ($p < 0.05$; Fig. 2b, d; Table 1).

At the seed-dispersal and over-winter stages, seeds were only destroyed by rodents, whereas at seedling-emergence stage, 44.4% and 41.5% of seeds that did not become surviving seedlings at the end of experiments each year died from animal attack (primarily by rodents) and pathogen infestation, respectively (Supplementary Fig. 1b). We, therefore, examined whether directions and strengths of distance-dependent mortality induced by these two causes differed significantly among fragment types using GLMMs (see Methods). The proportions of dead seeds and seedlings due to both causes decreased with increasing DTNCT (i.e., NDisDM) in all fragment types ($p < 0.05$ in all tests except the seed/seedling mortality rate by pathogen infestation on the medium islands; Table 2) except the small islands. On the small islands, there was significant PDisDM in proportions of dead seeds and seedlings caused by pathogen infestation and NDisDM formed by animal attack ($p < 0.05$ in both tests; see Table 2), indicating non-negligible contribution of

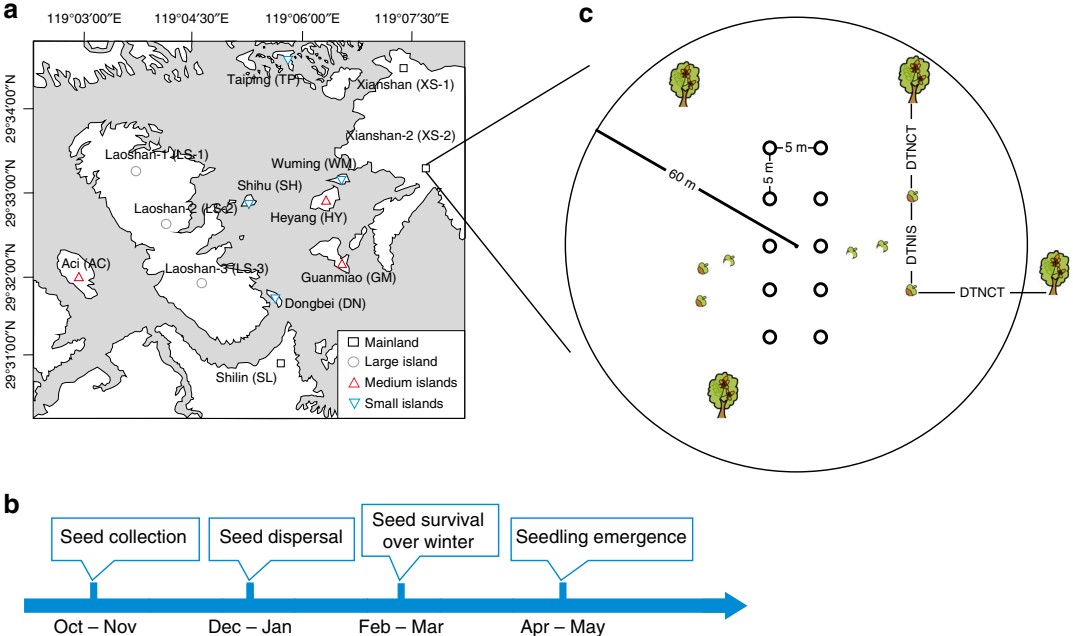

**Fig. 1** Locations of the experimental sites (**a**), timeline (**b**), and experiment design for the seed–seedling transition experiment (**c**). **b** After seed collection in October and November, the seed–seedling transition experiment was carried out from December to May of the next year, comprising the seed-dispersal, over-winter, and seedling-emergence stages. **c** In each experimental site, we set up two parallel transects that were about 5 m apart and placed five seed stations along each transect with an interval of c. 5 m; searching radius was 60 m from the geometric center of seed stations; at each seed–seedling transition stage, we measured the distance to the nearest conspecific tree (DTNCT) for each tracked seed and the distance to the nearest intact seed/seedling (DTNIS) for each intact seed/surviving seedling

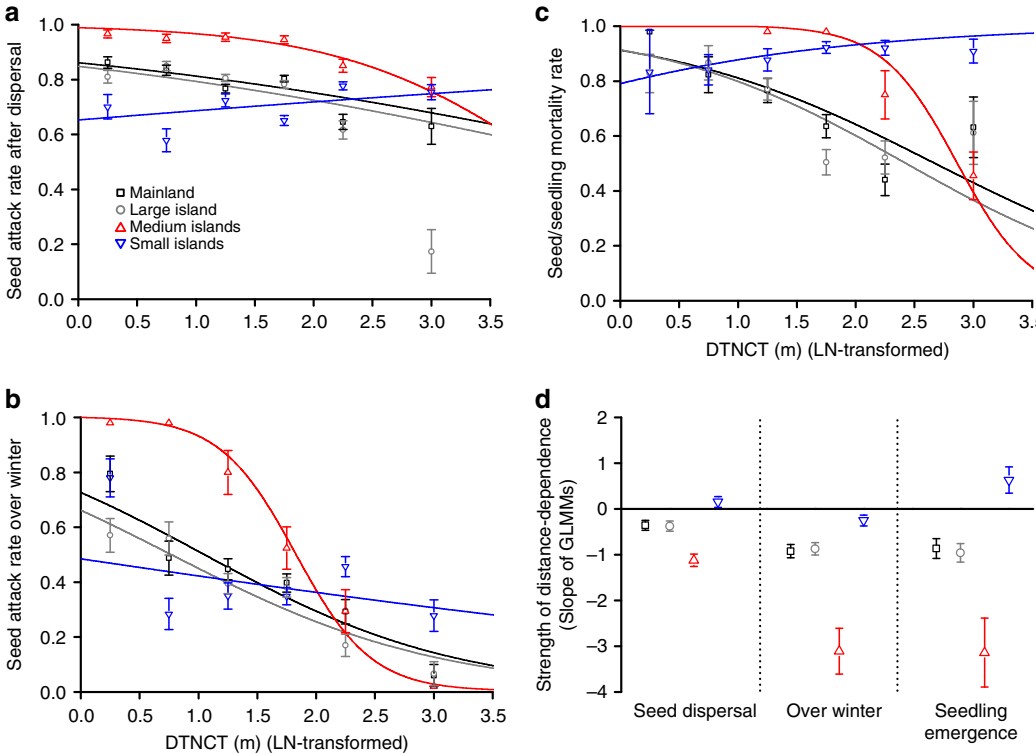

**Fig. 2** Trends of seed attack rate or seed/seedling mortality rate towards increasing DTNCT in all fragment types at each seed–seedling transition stage (**a–c**), and strengths of distance-dependence reflected by the slopes of GLMMs (mean (±S.E.)) at different stages (**d**). **a–c** Seed attack rates or seed/seedling mortality rates were calculated and shown for every 0.5 unit of log-transformed DTNCT, except the first and the last one, which included all seeds with LN-transformed DTNCTs smaller than 0.5 (<1.65 m) (shown at 0.25) and larger than 2.5 (>12.18 m) (shown at 3.00), respectively. We tested the differences in strengths of distance-dependence among different fragment types at different stages using GLMMs, and results were shown in Table 1

**Table 1 Comparisons of the relationships between the seed attack rate or the seed/seedling mortality rate and DTNCT among different fragment types at each seed–seedling transition stage, using GLMMs assuming binomial distribution of residuals. *P*-values of all tests were shown and Bonferroni correction was used to evaluate the significance for all pair-wise tests**

| Response variable | Fixed effects | Model parameters | | LR tests | | Z-tests | |
|---|---|---|---|---|---|---|---|
| | | Residual deviance | df of residuals | df | LR (*p*-value) | Pair-wise comparisons (slope (mean ± S.E.)) | z-value (*p*-value) |
| Seed attack rate after dispersal | Fragment type × DTNCT (LN-transformed) | 9637.8 | 10248 | 3 | 82.86 (<0.001)*** | Mainland (−0.36 ± 0.07***) vs. large island (−0.38 ± 0.08***) | 0.16 (0.876)[NS] |
| | | | | | | Mainland vs. medium islands (−1.12 ± 0.14***) | 4.91 (<0.001)*** |
| | | | | | | Mainland vs. small islands (0.15 ± 0.07*) | −5.06 (<0.001)*** |
| | | | | | | Large island vs. medium islands | 4.69 (<0.001)*** |
| | | | | | | Large island vs. small islands | −4.97 (<0.001)*** |
| | | | | | | Medium islands vs. small islands | −8.39 (<0.001)*** |
| Seed attack rate over-winter | Fragment type × DTNCT (LN-transformed) | 2561.7 | 2039 | 3 | 59.81 (<0.001)*** | Mainland (−0.92 ± 0.15***) vs. large island (−0.87 ± 0.14***) | −0.27 (0.786)[NS] |
| | | | | | | Mainland vs. medium islands (−3.11 ± 0.50***) | 4.20 (<0.001)*** |
| | | | | | | Mainland vs. small islands (−0.25 ± 0.12*) | −3.58 (<0.001)** |
| | | | | | | Large island vs. medium islands | 4.33 (<0.001)*** |
| | | | | | | Large island vs. small islands | −3.44 (0.001)** |
| | | | | | | Medium islands vs. small islands | −5.57 (<0.001)*** |
| Seed/seedling mortality rate | Fragment type × DTNCT (LN-transformed) | 1215.1 | 1215 | 3 | 39.56 (<0.001)*** | Mainland (−0.87 ± 0.21***) vs. large island (−0.96 ± 0.20***) | 0.32 (0.751)[NS] |
| | | | | | | Mainland vs. medium islands (−3.14 ± 0.75***) | 2.90 (0.004)* |
| | | | | | | Mainland vs. small islands (0.63 ± 0.29***) | −4.19 (<0.001)*** |
| | | | | | | Large island vs. medium islands | 2.80 (0.005)* |
| | | | | | | Large island vs. small islands | −4.53 (<0.001)*** |
| | | | | | | Medium islands vs. small islands | −4.68 (<0.001)*** |

[NS]Not significant, *p < 0.05, **p < 0.01, ***p < 0.001

pathogen infestation to overall PDisDM of the seed/seedling mortality rate.

In addition to these spatial signatures of distance-dependence, we also observed a temporal change of seed/seedling density curves during the experiment. The peak of intact seed/surviving seedling density shifted away from the nearest conspecific tree on the mainland and large and medium islands (Fig. 3a–c). This shift was clearest on the medium islands where NDisDM was also strongest (Fig. 3c); however, there was an opposite shift on the small islands (Fig. 3d).

We also tested if the extent of spatial aggregation of intact seeds/surviving seedlings changed among fragment types at each seed–seedling transition stage by assessing the difference in the distance from an intact seed to the nearest intact seed (DTNIS) (see Fig. 1c and Methods) using linear mixed models (LMMs)[26,27]. We found that DTNIS was significantly greater on the medium islands than other fragment types at each stage (p < 0.001 in all tests; Supplementary Fig. 1c, Table 3), indicating a more scattered pattern of intact seeds/seedlings, and there were no significant differences among other fragment types (p > 0.05 in all tests; Supplementary Fig. 1c, Table 3). These suggested that, compared with larger habitat fragments, DTNIS-dependent seed predation was weakened and consistent on the medium and small islands, respectively, and thus was unlikely to influence the pattern of DTNCT-dependent mortality in each fragment type.

**Effects of rodents**. We conducted capture-recapture experiments (see Methods) to investigate rodent density at each site. We caught a total of 512 rodents belonging to two rat species (*Niviventer fulvescens* and *N. confucianus*) during our 4-year study, but no marked rodents were recaptured. Over the 4 years, mean rodent density was 0.26 ± 0.02 individuals/trap/night on the medium islands, which was at least 3.7 times higher than that for other fragment types (LMMs: p < 0.001 in all tests; see Table 3); however, there were no significant differences among the other three fragment types (LMMs: p > 0.05 in all tests; Table 3). Furthermore, we evaluated the effects of rodent density on the slope of the linear relationship between seed attack rate or seed/seedling mortality rate and DTNCT, i.e., the strength of distance-dependent mortality (slope was denoted as SGAD) at each experimental site using LMMs. Rodent density had a significant negative impact on SGAD at all seed–seedling transition stages (Table 4), indicating its crucial role in shaping distance-dependent mortality.

## Discussion

Our results showed that both direction and strength of distance-dependent regeneration of *C. sclerophylla* varied with fragment size, suggesting its non-negligible role in shaping the spatial distribution of local dominant plants. In contrast to the NDisDM

**Table 2 Comparisons of the relationships between proportion of dead seeds and seedlings caused by animal attack/pathogen infestation towards increasing DTNCT among different fragment types at the seedling-emergence stage, using GLMMs assuming binomial distribution of residuals. *P*-values of all tests were shown and Bonferroni correction was used to evaluate the significance for all pair-wise tests**

| Response variable | Fixed effects | Model parameters | | LR tests | | Z-tests | |
|---|---|---|---|---|---|---|---|
| | | Residual deviance | df of residuals | df | LR (*p*-value) | Pair-wise comparisons (slope (mean ± S.E.)) | z-value (*p*-value) |
| Proportion of dead seeds and seedlings caused by animal attack | Fragment type × DTNCT (LN-transformed) | 1479.5 | 1215 | 3 | 6.90 (0.075)[NS] | Mainland (−0.40 ± 0.20*) vs. large island (−0.43 ± 0.18*) | 0.11(0.915)[NS] |
| | | | | | | Mainland vs. medium islands (−1.65 ± 0.49***) | 2.39 (0.017)[NS] |
| | | | | | | Mainland vs. small islands (−0.51 ± 0.20*) | 0.39 (0.698)[NS] |
| | | | | | | Large island vs. medium islands | 2.38 (0.017)[NS] |
| | | | | | | Large island vs. small islands | 0.30 (0.765)[NS] |
| | | | | | | Medium islands vs. small islands | −2.19 (0.028)[NS] |
| Proportion of dead seeds and seedlings caused by pathogen infestation | Fragment type × DTNCT (LN-transformed) | 1298.1 | 1215 | 3 | 28.69 (<0.001)*** | Mainland (−0.85 ± 0.25***) vs. large island (−0.66 ± 0.20**) | −0.61 (0.542)[NS] |
| | | | | | | Mainland vs. medium islands (−1.72 ± 0.94[NS]) | 0.90 (0.366)[NS] |
| | | | | | | Mainland vs. small islands (0.46 ± 0.18*) | −4.25 (<0.001)*** |
| | | | | | | Large island vs. medium islands | 1.12 (0.263)[NS] |
| | | | | | | Large island vs. small islands | −4.11 (<0.001)*** |
| | | | | | | Medium islands vs. small islands | −2.28 (0.022)[NS] |

[NS]Not significant, *p < 0.05, **p < 0.01, ***p < 0.001

in larger habitat fragments, we detected a drastic change on small islands (1.1–3.9 ha), where significant PDisDM occurred in seed-seedling transition. Such alteration subsequently resulted in aggregation of seeds/seedlings nearby conspecific trees on small islands, but inverse patterns were evident in other fragment types. In addition, compared with NDisDM strength on the mainland (>300 ha) and the large island (875 ha), much greater NDisDM was observed on medium islands (13–51 ha). As proposed in some previous studies[8,10], our results likely reflect changes in the relative importance of biotic and abiotic factors in habitat fragments of varying sizes.

Reduced fragment size and subsequent degradation of abiotic environments (i.e., stronger environment filtering) are the most likely causes forming the PDisDM on small islands. Small fragment size often triggers intensive edge effects, which may have extended into most of the area and substantially changed factors of the abiotic environment such as temperature and humidity in the forest interior[13,15,28]. The minimum distances from forest center to the edge on small islands were only c. 40–100 m, indicating that edge effects entirely penetrated these islands. Given that, only seeds/seedlings close to large *C. sclerophylla* trees would benefit from the favorable environment, e.g., cooler air and soil temperatures and higher moisture. Natural enemies like rodents also appeared to be seriously disturbed by such environment filtering, probably because they are at higher trophic levels compared to plants[11,20,21]. Consequently, the restricted rodent density on the small islands strongly impaired the strength of NDisDM, and the preference of some pathogens in the edge zone might lead to severe seed/seedling mortality[25], facilitating the formation of PDisDM at the seedling-emergence stage.

As fragment size increased, the dominance of abiotic environments in the distance-dependent regeneration tended to be replaced with biotic factors, largely due to the weakened edge effects. On medium islands, the strengthened NDisDM resulted from elevated seed/seedling mortality nearby large conspecifics, probably driven by increased densities of plants' natural enemies[6,7]. Rodents were the predominant natural enemies of *C. sclerophylla* seeds, and their densities were a significant driving force shaping the strength of distance-dependent seed/seedling mortality at all seed-seedling transition stages. This reflected a heterogeneous spatial distribution of rodents, given that well-developed roots and large crop sizes from big nut-bearing trees provide preferable microhabitats for rodents[29]. Moreover, given the strong preference of rodents to *C. sclerophylla* seeds (Supplementary Table 1), the elevated rodent density on medium islands, likely stimulated by absence of top predators[11,17], was assumed to mainly appear nearby adult *C. sclerophylla* trees rather than places lacking food resources, thus aggravating seed predation in the vicinity of these trees and strengthening NDisDM.

Infestation by pathogens also played an important role in plant regeneration success[6,25,27]. Large trunks and thick leaf litters supply rich resources that can support a mass of host-specific pathogens that can infest seeds/seedlings, resulting in higher mortality of seeds/seedlings nearby conspecific trees[3,5,30–32]. In accordance with these studies, at the seedling-emergence stage, we observed significant NDisDM induced by pathogens in all fragment types except small islands potentially due to altered microclimates[33–35], suggesting that environments in large- and medium-sized forests could sustain the aggregation of pathogens close to large trees.

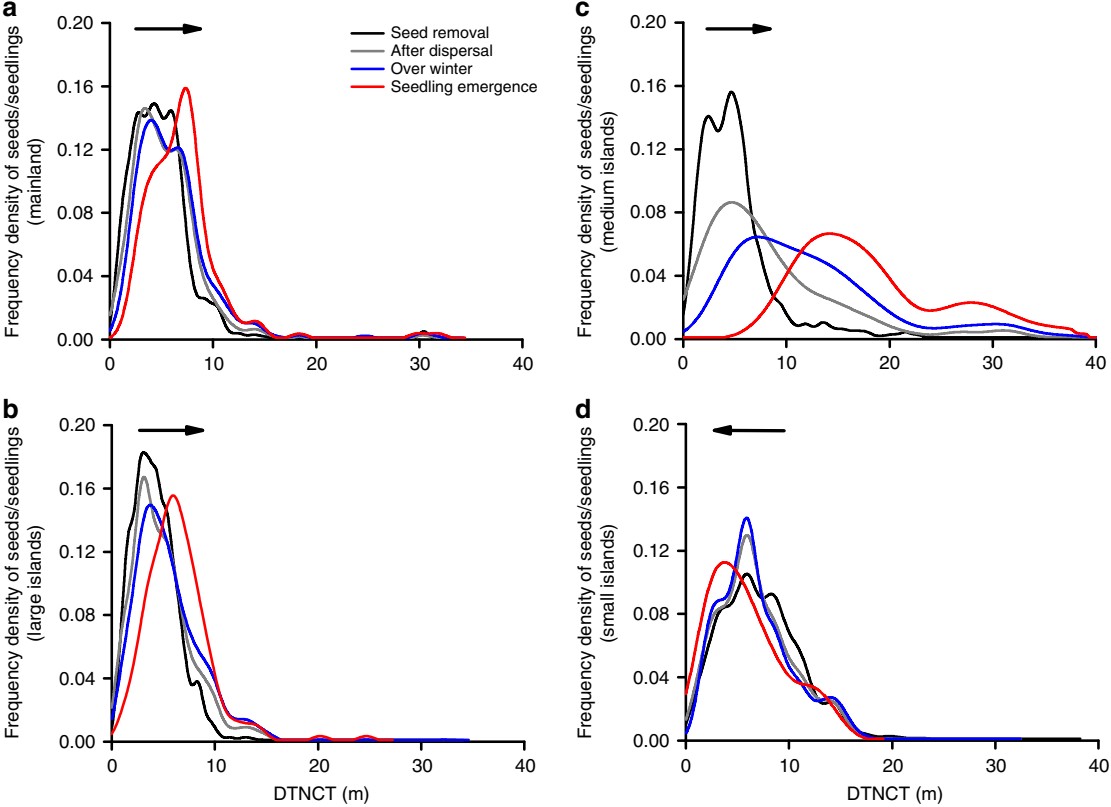

**Fig. 3** Frequency density distributions of seeds/seedlings with increasing DTNCT at different seed–seedling transition stages in each fragment type (**a–d**). Black, gray, blue, and red curves represent the frequency density distributions of the removed seeds (containing both the intact and destroyed seeds at the seed-dispersal stage), intact seeds after dispersal and over-winter, and seeds that became surviving seedlings by the end of seed–seedling transition experiments, respectively

In addition to fragment size, density of adult conspecific trees and degree of spatial aggregation of conspecific seeds/seedlings can also influence density-dependent predation and change the strength of distance-dependence[9,36]. Nevertheless, these factors were unlikely to affect our results because we found similar densities of large *C. sclerophylla* trees in all fragment types (see Methods), and densities of seeds/seedlings (reflected by DTNIS) on islands were not higher than on the mainland. In addition, survival of plant propagules may also be affected by nearby heterospecifics[2,7], which needs to be tested in future studies; however, their effects were likely to be limited given that *C. sclerophylla* was the dominant species (see Methods).

Based on our results, we speculate that the relationship between the strength of distance-dependent mortality of *C. sclerophylla* seeds and fragment size would fit a unimodal curve, with a turning point from which NDisDM will switch to PDisDM when fragment size falls below some threshold (Fig. 4). The switch from NDisDM to PDisDM is likely to have dramatic adverse impacts on local biodiversity on small islands in the TIL because, in contrast to NDisDM that facilitates species coexistence[2,7], PDisDM likely restricts the regeneration of other plant species via depriving the chances of occupying resource-rich microhabitats despite seeds of these species being less likely to be attacked by rodents (see Methods), thus accelerating species extinction and generating extinction debt in extremely small forest patches. Moreover, the generally weakened plant-enemy interaction may cause strong interspecific competition and decrease plant diversity in edge zones[20,33]. However, such degradation in biodiversity may be alleviated because some shade-intolerant species prefer edge zones[37]. Considering the uncertain impacts of PDisDM likely caused by environment

filtering on current biodiversity loss in highly fragmented forests, future studies need to verify whether this relationship is truly unimodal using more fragment sizes to fill the size gap between fragment types and should include more plant species with diverse life history characteristics. Additionally, the strengthened NDisDM in medium-sized forests may benefit other plant species, especially those with different life history traits, and contribute to the maintenance of biodiversity.

Changes in global climate and land use are increasingly aggravating habitat fragmentation worldwide, separating remnant forests into further smaller patches and threatening biodiversity[13,15,38,39]. Assessing species loss in fragmented forests is crucial to conservation and restoration design[40] but the extinction rates in small remnant forests are highly likely to be underestimated, because none of the existing models such as species–area relationships[41] consider non-neutral processes such as fragment size-dependent alteration in distance-dependent regeneration of dominant plant species. More case studies are therefore urgently needed to test whether similar findings can be detected in other fragmented ecosystems and improve the existing island biogeography frameworks, for more precise predictions of the consequences of habitat fragmentation.

## Methods

**Study area and the focal tree species**. Our study was carried out in the southeastern Thousand-Island Lake (TIL), a typical land-bridge island system formed by Xin'an Jiang Dam in 1959, in Zhejiang Province, China. Land-bridge island systems created by hydroelectric dams provide a large number of forest patches with a series of sizes and clear boundaries, and so often serve as "natural laboratories" for testing ecological and evolutionary theories[15].

The TIL contains more than 1000 islands varying greatly in size[42]. *Castanopsis sclerophylla* (Fagaceae), a monoecious nut-bearing tree, is one of the dominant trees in

**Table 3 Comparisons of DTNIS at each seed–seedling transition stage and the rodent density (individual/trap/night) among different fragment types using LMMs and GLMMs assuming binomial distribution of residuals (with residual deviance and df of residuals). P-values of all tests were shown and Bonferroni correction was used to evaluate the significance for all pairwise tests**

| Response variable | Fixed effect | Models | LR tests | | T/Z-tests | | |
|---|---|---|---|---|---|---|---|
| | | | df | LR (p-value) | Pair-wise comparisons (mean ± S.E.) | df | t/z-value (p-value) |
| DTNIS after dispersal (LN-transformed) | Fragment type | LMMs | 3 | 20.07 (<0.001)*** | Mainland vs. large island | 9 | 0.11 (0.913)NS |
| | | | | | Mainland vs. medium islands | 11 | −5.49 (<0.001)** |
| | | | | | Mainland vs. small islands | 9 | 1.30 (0.223)NS |
| | | | | | Large island vs. medium islands | 9 | −4.15 (0.002)* |
| | | | | | Large island vs. small islands | 9 | 0.79 (0.449)NS |
| | | | | | Medium islands vs. small islands | 11 | 7.05 (<0.001)*** |
| DTNIS over-winter (LN-transformed) | Fragment type | LMMs | 3 | 19.27 (<0.001)*** | Mainland vs. large island | 7 | 0.42 (0.685)NS |
| | | | | | Mainland vs. medium islands | 11 | −5.19 (<0.001)** |
| | | | | | Mainland vs. small islands | 9 | 1.29 (0.229)NS |
| | | | | | Large island vs. medium islands | 8 | −4.38 (0.003)* |
| | | | | | Large island vs. small islands | 7 | 0.50 (0.636)NS |
| | | | | | Medium islands vs. small islands | 12 | 6.70 (<0.001)*** |
| DTNIS seedling-emergence (LN-transformed) | Fragment type | LMMs | 3 | 24.38 (<0.001)*** | Mainland vs. large island | 6 | 1.75 (0.128)NS |
| | | | | | Mainland vs. medium islands | 45 | −6.73 (<0.001)*** |
| | | | | | Mainland vs. small islands | 26 | −0.75 (0.461)NS |
| | | | | | Large island vs. medium islands | 45 | −7.79 (<0.001)*** |
| | | | | | Large island vs. small islands | 26 | −2.02 (0.053)NS |
| | | | | | Medium islands vs. small islands | 86 | 5.36 (<0.001)*** |
| Rodent density | Fragment type | GLMMs (2893, 4581) | 3 | 17.69 (<0.001)*** | Mainland (0.070 ± 0.02) vs. large island (0.069 ± 0.01) | – | 0.01 (0.992)NS |
| | | | | | Mainland vs. medium islands (0.26 ± 0.02) | – | −5.01 (<0.001)*** |
| | | | | | Mainland vs. small islands (0.057 ± 0.01) | – | 1.01 (0.313)NS |
| | | | | | Large island vs. medium islands | – | −3.74 (<0.001)** |
| | | | | | Large island vs. small islands | – | 0.74 (0.459)NS |
| | | | | | Medium islands vs. small islands | – | 6.28 (<0.001)*** |

NSNot significant, *p < 0.05, **p < 0.01, ***p < 0.001

**Table 4 Effects of rodent density on the slope of the generalized linear regression between seed attack rate or seed/seedling mortality rate and DTNCT (SGAD) at different seed–seedling transition stages using LMMs**

| Response variable | Fixed effect | Slope (mean ± S.E. (N samples)) | LR tests | |
|---|---|---|---|---|
| | | | df | LR (p-value) |
| SGAD (seed-dispersal) | Rodent density | −0.862 ± 0.258 (51) | 1 | 7.81 (0.005)** |
| SGAD (over-winter) | Rodent density | −1.652 ± 0.313 (50) | 1 | 13.36 (<0.001)*** |
| SGAD (seedling-emergence) | Rodent density | −1.743 ± 0.587 (48) | 1 | 7.64 (0.006)** |

LR tests likelihood ratio tests
**p < 0.01, ***p < 0.001

most of the islands in the southeast TIL[43,44]. This species flowers during April–May and produces large amounts of mature seeds during October–November. Early studies showed that some parasitic insects, mainly *Curculio* weevils, are the specific pre-dispersal seed predators[45,46] and that rodents are the primary seed consumer and primary mode of seed-dispersal[40]. Therefore, NDisDM may function in regulating the population size of *C. sclerophylla* in TIL. This prior knowledge was the main reason for choosing this local dominant species for study. Additionally, *C. sclerophylla* is common in other Chinese subtropical evergreen broadleaved forests[44,47]; thus, insights from this study may have general application in other subtropical evergreen forests.

**Selection of fragments with different fragment sizes**. To study the effect of fragment size on the distance-dependent regeneration of *C. sclerophylla*, we defined four types of habitat fragments according to the distribution of island sizes in our

study area (Fig. 1a): mainland — three large forest patches (XS1, XS2, and SL, >300 ha); large island — the only large island (LS, 875 ha); medium islands — three medium-sized islands (AC, GM, and HY, 13-51 ha); and small islands — four small islands (DN, SH, TP, and WM, 1.1-3.9 ha). Three experimental sites with a minimum interval of 2 km were established on the large island, and one experimental site was set on each of the selected mainland forest patches and medium and small islands (Fig. 1a).

In each experimental site, *C. sclerophylla* was the most dominant tree species with a mean relative abundance of 0.68, which did not vary significantly among different fragment types (Supplementary Table 1); the average relative abundance of the second dominant species, *Pinus massoniana*, was 0.17 and its seeds are inedible for rodents. Furthermore, we found a significant feeding preference of rodents for *C. sclerophylla* seeds in our pre-experiment. When we mixed seeds of this species and other two common Fagaceae species (*Lithocarpus glaber* and *Quercus glandulifera*) in the TIL (200 seeds for each species), *C. sclerophylla* seeds

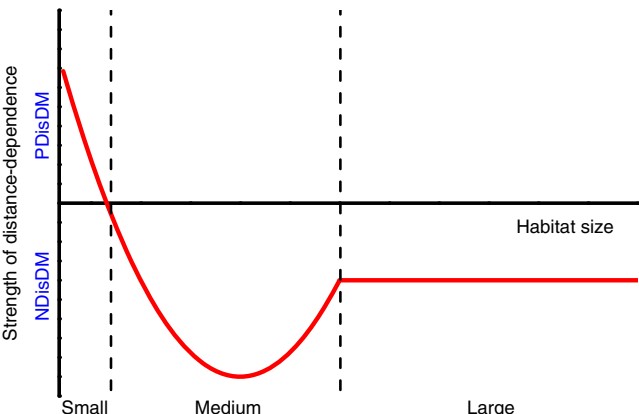

**Fig. 4** Schematic representation of how the direction (negative (NDisDM) or positive (PDisDM) distance-dependence) and strength of distance-dependent mortality vary with fragment size

suffered far higher attack rates by rodents (97.0%) than the other two species (23.5% and 31.5% for *L. glaber* and *Q. glandulifera*) (Supplementary Table 1). These suggest that the distance-dependent seed–seedling transition of *C. sclerophylla* greatly contributes to the spatial recruitment pattern of local plant communities and that rodents act as specific seed predators in our sites. In addition, population densities of *C. sclerophylla* trees (height > 1.5 m) did not vary significantly among experimental sites for the different fragment types (Supplementary Table 1), indicating that variation in densities of *C. sclerophylla* was unlikely to affect distance-dependent seed–seedling transition patterns.

**Seed–seedling transition experiment**. During 2009–2012, we repeated the seed–seedling transition experiment every year from autumn to the next spring in all experimental sites. In each year, the experiment started with the collection of *C. sclerophylla* seeds after October, and we then monitored the seed-dispersal, overwinter survival, and seedling-emergence until the end of the next spring (Fig. 1b). In the autumn of each year, intact *C. sclerophylla* seeds were collected from at least 50 adult trees in all experimental sites in the TIL. At least 10 days after the end of seed fall in each year (early December), ten seed stations, each covering a circular area of radius 0.25 m, were set up in each experimental site (Fig. 1c). The reason to set ten seed stations in each experimental site was to keep the initial distances from placed seeds to the nearest conspecific trees (height > 1.5 m) (DTNCTs) as consistent as possible across all fragment types (the average DTNCT ranged from 4.34 to 4.78 m across four fragment types with no significant difference; see Supplementary Table 1), in case of any differences in starting distance-dependent conditions that could bias the seed–seedling transition pattern. Fifty seeds were placed in each seed station and 500 seeds per experimental site. Predation satiation is unlikely to occur to strongly influence the distance-dependent pattern with an annual placement of 500 seeds in each experimental site because, in 2009, we recorded the crop sizes of 327 *C. sclerophylla* trees in all experimental sites and found high seed production (1837 ± 322 seeds/ha; mean ± S.E.) with no significant differences among fragment types (Supplementary Table 1). Each seed was labeled by a light plastic tag with a unique serial number. The total weight of a tag was only 6.1% of the weight of a *C. sclerophylla* seed (1.234 ± 0.014 g; *n* = 1509) and therefore was assumed to have no significant impact on seed-dispersal. Note that the seed–seedling transition data were missing on the small island TP in 2009 because all seed stations on this island were destroyed by a fallen tree.

Each day after the initial placement of seeds in seed stations, we checked the number of seeds remaining in each seed station, and carefully searched the seeds dispersed away until all seeds were removed or eaten in each experimental site (Fig. 1c). All dispersed seeds were assigned into three categories: intact (intact seeds that were shallowly buried in soil or leaf litter), attacked (seeds that were attacked by animals with only husks and tags left), and missing. Missing seeds were unlikely to contribute to seedling recruitment because they were largely hoarded in rat holes far below ground. In spring 2010, we dug all rat holes within an experimental site on the large island and found 91% of the missing seeds. This number also suggested that the estimation of each category of dispersed seeds was sufficiently accurate. All intact and destroyed seeds after dispersal were mapped, and we measured their DTNCTs and the distance from an intact seed to the nearest intact seed (DTNIS) (Fig. 1c).

We conducted the field survey in early spring of the next year to track the intact seeds after dispersal and record their status over-winter: intact, attacked, or missing. In the following late spring and early summer, we checked whether an intact seed became a surviving seedling. Unlike the former two stages, in which all destroyed seeds were only due to animal attack, failure of seeds to become surviving seedlings could be attributed to three causes: animal attack — seeds/seedlings attacked by animals, primarily rodents and sometimes wild boars, with

only husks/seedling remains (with tags) left; pathogen infestation — rotten seeds/seedlings with mildew appearance; and un-germinated seeds — seeds that remained intact but did not germinate by the end of May. The DTNCT and the DTNIS of each intact seed/surviving seedling at these two stages were determined in each experimental site.

**Rodent capture–recapture experiment**. We repeatedly surveyed the rodent density after the seed-dispersal stage in the seed–seedling transition experiment (in January) during 2010–2013. In each experimental site, we placed 30 rodent cages along three 90-m transects (10 cages in each transect) with 10-m intervals for three consecutive nights. We checked all cages daily to mark and morphologically identify the taxa of captured rodents and then released them. However, no marked rodents were recaptured during our four years of experiments, and we therefore calculated the proportion of cages that captured rodents per night as the estimate of rodent density at each site.

**Data analyses**. To test whether strength of distance-dependent mortality (represented by the regression slopes between seed attack rate or seed/seedling mortality rate and DTNCT) at each seed–seedling transition stage varied among different fragment types, we evaluated the effects of interaction between fragment type and DTNCT on seed attack rate or seed/seedling mortality rate using GLMMs implemented in package *lme4* version 1.1-12[26] in R 3.4.4[48], assuming binomial distribution of residuals. We used likelihood ratio (LR) tests to examine the significance of interaction by comparing two GLMM formulas in R language formats: (1) glmer(seed attack rate or seed/seedling mortality rate ~ fragment type × LN(DTNCT) + year + (1|island/experimental site), family = 'binomial'), and (2) glmer(seed attack rate or seed/seedling mortality rate ~ fragment type + LN(DTNCT) + year + (1|island/experimental site), family = 'binomial'), where LN represents the natural logarithm. According to the design of our seed–seedling transition experiment, we set experimental sites nested in islands (i.e., three experimental sites on the large island, and the mainland forest patches were also included as specific sites) as the random effect. Based on GLMM formula (1), we then quantified and compared the strength of distance dependence at different seed–seedling transition stages using Z-tests with Bonferroni corrections.

We also examined whether fragment size (sizes of islands and mainland forest patches) significantly influenced strength of distance-dependent mortality at each seed–seedling transition stage using LR tests to compare two GLMM formulas: glmer(seed attack rate or seed/seedling mortality rate ~ LN(fragment size) × LN(DTNCT) + year + (1|island/experimental site), family = 'binomial'), and glmer (seed attack rate or seed/seedling mortality rate ~ LN(fragment size) + LN(DTNCT) + year + (1|island/experimental site), family = 'binomial'). The results showed that at each seed–seedling transition stage, the regression slopes between seed attack rate or seed/seedling mortality rate and DTNCT decreased with increasing fragment size (reflected by a significant negative interaction between fragment size and DTNCT) (see Supplementary Table 2). However, we only retained the results from the GLMMs using fragment type, because they displayed a non-linear relationship between the strength of distance-dependent mortality and fragments with different sizes (see Results and Fig. 2) and there was a huge gap in fragment size among fragment types (rather than a continuous distribution).

At seedling-emergence stage, both animal attack and infestation by pathogens contributed to seed/seedling mortality, and thus their responses to changes in fragment size could crucially influence distance-dependent seed–seedling transition. We therefore examined whether directions and strengths of distance-dependent mortality induced by these two causes (animal attack and pathogen infestation) differed significantly among fragment types using LR tests in GLMMs in *lme4* assuming binomial distribution of residuals and setting experimental sites nested in islands as the random effect. GLMM formulas for comparison were: (3) glmer(proportion of dead seeds and seedlings caused by animal attack (to all seeds that were intact at the beginning of seedling-emergence stage) ~ fragment type × LN(DTNCT) + year + (1|island/experimental site), family = 'binomial'), and (4) glmer(proportion of dead seeds and seedlings caused by animal attack ~ fragment type + LN(DTNCT) + year + (1|island/experimental site), family = 'binomial'). We then ran similar models for proportions of dead seeds and seedlings caused by pathogen infestation. Strengths (slopes in GLMM formula (3)) of distance-dependent mortality by these two causes were quantified and compared using Z-tests with Bonferroni corrections.

In addition, spatially aggregated conspecific seeds/seedlings were assumed to suffer higher mortality in the presence of density-dependent seed predation than scattered seeds/seedlings[9,33] and therefore may have influenced distance-dependence of seed–seedling transition. Hence, we tested if the extent of spatial aggregation of intact seeds/surviving seedlings changed among fragment types at each seed–seedling transition stage by assessing the difference in DTNIS (longer DTNIS indicates a more scattered pattern). Specifically, using LR tests to compare linear mixed models (LMMs) in *lme4*, we set experimental sites nested in islands as the random effect. LMM formulas for comparison were: (1) lmer(LN(DTNIS + 1) ~ fragment type + year + (1|island/experimental site)), and (2) lmer(LN(DTNIS + 1) ~ year + (1|island/experimental site)). Based on LMM formula (1), we conducted pair-wise comparisons of DTNIS between fragment types using *t*-tests with Bonferroni corrections. LR tests and *t*-tests were carried out in R package *lmerTest* version 3.0-1[27].

The difference in rodent density (represented by proportion of cages that captured rodents per night) among fragment types was examined by using LR tests to compare GLMMs in *lme4*, assuming binomial distribution of residuals (GLMM formulas for comparison: (5) glmer(rodent density ~ fragment type + year + (1| island/experimental site)), family = 'binomial'), and (6) glmer(rodent density ~ year + (1|island/experimental site)), family = 'binomial')). Pair-wise comparisons between fragment types were then conducted using Z-tests with Bonferroni corrections. Furthermore, to evaluate the role of rodents in shaping the distance-dependence in seed–seedling transition, we analyzed the effects of rodent density on the slope of the generalized linear relationship between seed attack rate or seed/ seedling mortality rate and DTNCT (the slope was denoted as SGAD) at each experimental site at each seed–seedling transition stage, using LR tests in LMMs in *lme4* and setting experimental sites nested in islands as the random effect. LMM formulas for comparison were: (3) lmer(SGAD ~ rodent density + year + (1|island/ experimental site)), and (4) lmer(SGAD ~ year + (1|island/experimental site)).

**Reporting summary**. Further information on research design is available in the Nature Research Reporting Summary linked to this article.

## Data availability
Data have been deposited in Dryad[49] (https://doi.org/10.5061/dryad.57q7865). Any other relevant data not available in the article or supplementary information are available from the authors on reasonable request.

## Code availability
All codes for data analysis are provided in Methods.

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

## Acknowledgements

We thank Bin Ai, Yuan Miao, and Qian Zhang for their kind help in field experiments, and Spencer Barrett for constructive suggestions in data analysis. The study was founded by NSFC grants 31361123001 and 30970430 to X.Y.C. and 31870404 to G.C.S. and NSF grants DEB-1342751 and DEB-1342757 to X.Y.C. and was supported by ECNU Multi-functional Platform for Innovation (008).

## Author contribution

R.W., X.Y.C. and G.C.S. designed the experiments and analyzed data, R.W., Y.S.S., Y.X.Z. and G.F.X. collected the data, and all authors contributed to manuscript writing.

## Additional information

**Competing interests:** The authors declare no competing interests.

