## [Peer Review File · Communications Biology]

Editorial Note: This manuscript has been previously reviewed at another journal that is not operating a transparent peer review scheme. This document only contains reviewer comments and rebuttal letters for versions considered at *Communications Biology*.

Reviewers' comments:

Reviewer #1 (Remarks to the Author):

The authors have made some changes in line with the comments of myself and the other reviewers. This continues to be a paper on an important and interesting topic. However, the analyses still require some modification before they can be used as a basis for inference. In particular, the random effects structure of the model does not align with the hypotheses being tested (e.g. why is island size-class a random intercept?). I have provided some advice below, but I would recommend the authors consult with colleagues with experience fitting mixed-effects models in R (and specifically the lme4 package) to work out the issues with these models.

The study tackles an important topic although the generality of the conclusions are somewhat limited by (1) use of a single species; (2) the potential role of habitat filtering in driving the positive density dependence on small islands; (3) the treatment of island size as a discrete predictor. None of these issues is a big problem individually, but together they do limit any extrapolations to other species or systems. Again, this does not make the manuscript uninteresting – but it might be a good idea for the authors to acknowledge this limitation more explicitly in the discussion.

Finally, the language could be polished – there are a large number of awkward sentences scattered throughout the manuscript (some examples in the detailed comments). This is coupled with several arguments that appear simplistic or rely on references that did not appear to support the specific points being made. Together, these problems made the manuscript a difficult read and should be improved prior to publication.

Detailed comments

L30: change to occur at high density.

L44: There is a distinction between increased seed/seedling predation and increased distance dependence. For example, increased predator densities could make it harder for seeds to escape from the area around parent trees, reducing NDisD. Although the authors qualify the statement with “potentially”, that does not change the fact that the argument is simplistic. The logic in the sentence about low species richness, L45-47 is also not very clear. Why should low species richness reduce distance dependence? It could potentially work in the other direction – for example, few species could support specialization on the remaining species (e.g. May 1991 argues you’d expect lower specialization by fungi on diverse host communities).

L186: Were these fungal pathogens actually isolated from seeds and seedlings in this system? The references cited don’t appear to mention these particular fungal pathogens (or really identify the fungal pathogens to genus). They also cover different plant species (although Liu et al 2012 considers two other *Castanopsis* species).

L152: The argument for the effect of island size (and edge effects) appears to hinge on *C. sclerophylla* being edge-sensitive. Because adults are indicative of non-edge habitats, recruitment is concentrated in these areas when the forest is dominated by edges. That argument seems to indicate that we are

not seeing "positive density dependence" as the authors describe it, but habitat filtering – which is a very different process and would not affect other species in the same way (e.g. an edge preferring species might end up with a pattern consistent with "negative distance dependence" because they were concentrated near the edge and away from existing conspecific adults). The outcome for biodiversity is also likely to be very different – because the habitat filtering mechanism doesn't lead to a rare species advantage, there is no predictable impact on biodiversity without knowing about the other species in the community.

L263: Doesn't follow that a preference of rodents for *C. sclerophylla* will lead to distance dependence. Yes, it makes it more likely, but distance dependence will also depend on (1) the foraging behavior of the rodents; (2) the distribution of rodent territories. There is certainly a possibility that distance dependence could be a factor here, but stating that distance dependence dominates spatial recruitment is an overreach.

Following on from my comment on L263 - It would also be helpful if the authors were more specific about the strength of the preference - "far higher" is a matter of interpretation – can you attach numbers to this preference so that they are not just in the supplement? An analysis of variance examining the proportion of the variance in predation attributed to host species might work as a means to quantify the specialization (or some measure from the host-specificity literature).

L334: The formulae for the binomial models don't seem to make much sense to me – why is island size category being treated as a random effect? Given that the test is of the null hypothesis that island size has no effect on survival and the effect of distance on survival, they should be fixed effects, not random effects. It is also a little strange to have habitat type in both fixed and random intercepts. Those two terms will be almost unidentifiable (the only reason they will be partially identifiable is because of the normality constraint on the random intercepts). Additionally, what is the distinction between habitat type and experimental site size? It appears that this formulation might have been in response to my comments on the last version – however, this is not quite what I was suggesting. A formulation like

```
glmer(cbind(deaths, survs) ~ island_size * log(dtnct) + (1|year) + (1|island/experimental_site),  
family=binomial)
```

Might be more appropriate.

Additionally, it is the author's prerogative to decide whether to use island size as a continuous or discrete predictor. However, the former is more generalizable and does not create artificial bins with the associated idiosyncrasies (e.g. a step difference between islands of size 10 ha and 11 ha). It is not necessarily true that interactions between continuous predictors are very hard to explain. It could be argued that it is far harder to explain 3 interactions (one with each size class) than a single one. The point about non-linearities is valid, but it should be acknowledged that the authors are imposing a specific non-linear structure on their model (i.e. one that has step changes between size-classes). The consequence is that it is very difficult to generalize from the result to other contexts. For example, is a small island in their experimental system equivalent to a small island in another experiment? It is far easier to make such comparisons when size is continuous.

Reviewer #3 (Remarks to the Author):

I reviewed the previous manuscript by Wang et al. As I said before, I am very impressed with the large amount of field work the authors undertook for this study, and I think the research questions are very interesting. My primary concerns were with the model structures used. While the authors have adequately addressed some of my previous concerns, I still do not think they are using the appropriate models in some cases. I have described my concerns below. I think if the authors can address these concerns, this would be a publishable study that would be of interest to many

ecologists.

The explanation of why year was kept as a random effect does not make sense: "We however still chose to set year as a random effect because: (1) our purpose was to show the change of distance-dependent mortality on small and medium islands comparing with larger habitats rather than the variation among different year; and (2) it is very difficult to explain a three-way interaction." I was not suggesting to include any interactions with year, but simply to include it as a categorical fixed effect, rather than a random effect. It does not matter that you were not primarily interested in variation among years. The problem is that you should not include a random effect that has less than 5-6 groups/levels. In your case with only 3-4 years, it is not appropriate to treat year as a random effect. You may find this blog post useful: <https://dynamiccecolgy.wordpress.com/2015/11/04/is-it-a-fixed-or-random-effect/>. I also recommend consulting the book by Gelman and Hill entitled "Data Analysis Using Regression and Multilevel/Hierarchical Models".

Another issue with the current model structure is that it does not make sense to include as a random effect the term: (1|habitat type/experimental site size). I think this was done in response to one of the other reviewer's comments, but I think the authors have misinterpreted what that reviewer was saying. They were not saying you needed to include experimental site size as a random effect. I believe they were saying that you need to include (1|year) + (1|island/site), because island size (as well as other factors) varies at the whole island level so you need to include site nested within island as a random effect, NOT that you should additionally include experimental site size as a random effect. It does not make sense to include experimental site size as a random effect. Please remove that term (1|habitat type/experimental site size) from the models were it was included.

I am also concerned with the analysis to examine the cause of mortality. Specifically, cause is included as a predictor variable in the model of mortality rate, but this doesn't make sense. How can mortality rate vary with cause? Were there attacked individuals that survived? If not, I don't understand this. Also, where is habitat type in these models? The goal was to test whether the mortality rate by the two causes different between habitat types. I think the model that makes sense is proportion of deaths caused by animal attack $\sim \text{LN}(\text{DTNCT}) * \text{habitat}$ and then a similar model for proportion of deaths caused by pathogen attack. For example, proportion of deaths caused by animal attack means if 100 seedlings died, and 80 of those deaths were due to animals, the proportion would be 80/100.

Other suggestions:

I think the term 'habitat type' and 'habitat size' should be changed to 'fragment size' throughout. The word habitat suggests some difference beyond size (e.g. in topography or soil nutrients).

There have been some very recent papers that might be relevant to mention and cite: Viswanathan et al. 2019 Proc Roy Soc B and Krishnadas et al. 2018 Nature Communications. Both examine effects of fragmentation on plant-pathogen interactions and the second one also looks at CNDD.

Since the Methods section comes after Results in this journal format, you need to explain the tests better here to help readers understand what you did, i.e. we examined .. and found...

Overall, the writing is very good. There are some small grammatical mistakes throughout the manuscript (which is completely understandable, in my opinion). I have made some suggested edits below:

Line 20 – change "strengthened" to "stronger"

Line 21 – change “positive distance-dependence of mortality” to “positive distance-dependent mortality”

Line 31 – add “pattern of” before mortality

Line 53 – change “frustrated” to “failed”

Line 54 – change “long-distance dispersed” to “dispersed long distances”

Line 61 – Check this recent paper

Line 63- Why are the sizes for medium and small islands still listed as 10-100 ha and <10 ha, when in the response to reviewers a narrower range was given (from 13 to 51 ha for medium and from 1.1 to 3.9 ha for small islands). Please use the more accurate narrower ranges.

Line 64 – What do you mean by “The relative and population densities”? By relative, do you mean relative abundance (ie relative to other species present)? If not, I don’t understand what ‘relative’ means here.

Line 68 – change to ‘four consecutive year’

Line 71 – The first question listed could be stated better. I suggest something like: “(1) Was there evidence of negative distance dependence at early life stages in this study system?”

Line 72 – “changed” should be ‘change’

Line 73 – add “did” after “how”

Line 74 – change “responded” to “respond”

Line 81 (and elsewhere) – change “could hardly” to “is unlikely to”

Line 82 (and elsewhere) – change “survival seedlings” to “surviving seedlings”

Line 99 – Does “animal attack” just mean rodents? Or also insects or other species?

Line 111 – change “drifted” to “shifted”

Line 133 – change “released” to “had”

Line 140 – I would remove or rephrase the statement “Although the factors influencing distance-dependent plant regeneration remain unclear”. It is not clear whether you are talking about your own study here or previous studies, and it is a confusing way to start the discussion.

Line 150 – change ‘must’ to ‘likely’

Line 153 – This argument suggests that overall seedling mortality should be higher on small islands than medium and large islands. Was this true?

Line 163 – “Higher trophic levels” compared to what?

Line 196 – What does ‘less effective’ mean here? Less effective at doing what?

Line 203 – change “must strongly restrict” to “likely restricts”

Line 210 – Remove the statement “as expected by the intermediate disturbance hypothesis”. This is not correct to say that the IDH predicts this and plus the IDH is somewhat controversial, so I don’t think you should mention it at all. It is not very relevant to your findings.

Line 239 – change ‘media’ to ‘mode’

Line 240 – change ‘priori’ to ‘prior’

Line 255 – again, I don’t understand what ‘relative density’ means? What are the units and how is this calculated?

Line 280 – Did you test whether initial DTNCTs of placed seeds was actually consistent across all habitat types?

Line 283 – You should write out numbers when it is the first word at the start of a sentence, so “Fifty” instead of “50”

Line 284 – change “may hardly occur” to “is unlikely to occur”

Line 300 – change “could hardly” to “are unlikely to”

Line 378-379 – I think by ‘rodent density’ you mean the proportion of traps with rodents, but rodent density per unit area. Please clarify because it does not make sense to use binomial errors unless your dependent variable is a proportion.

Responses to comments

Reviewers' comments:

Reviewer #1 (Remarks to the Author):

The authors have made some changes in line with the comments of myself and the other reviewers. This continues to be a paper on an important and interesting topic. However, the analyses still require some modification before they can be used as a basis for inference. In particular, the random effects structure of the model does not align with the hypotheses being tested (e.g. why is island size-class a random intercept?). I have provided some advice below, but I would recommend the authors consult with colleagues with experience fitting mixed-effects models in R (and specifically the lme4 package) to work out the issues with these models.

>>R: Thank you very much for your constructive comments on statistics, and we have revised models following your suggestions. Please see below for details.

The study tackles an important topic although the generality of the conclusions are somewhat limited by (1) use of a single species; (2) the potential role of habitat filtering in driving the positive density dependence on small islands; (3) the treatment of island size as a discrete predictor. None of these issues is a big problem individually, but together they do limit any extrapolations to other species or systems. Again, this does not make the manuscript uninteresting – but it might be a good idea for the authors to acknowledge this limitation more explicitly in the discussion.

>>R: Thanks! We agree that it is necessary for future studies to include more plant species and more islands with different sizes, as well as an explicitly investigation of environments in each island. We have explicitly pointed out these limitations in discussion at lines 186, 230-233 and 244-253.

Finally, the language could be polished – there are a large number of awkward sentences scattered throughout the manuscript (some examples in the detailed

comments). This is coupled with several arguments that appear simplistic or rely on references that did not appear to support the specific points being made. Together, these problems made the manuscript a difficult read and should be improved prior to publication.

>>R: We have revised the sentences according to reviewers' comments and invited an English native speaker to improve the English in our manuscript.

Detailed comments

L30: change to occur at high density.

>>R: Revised. Line 32.

L44: There is a distinction between increased seed/seedling predation and increased distance dependence. For example, increased predator densities could make it harder for seeds to escape from the area around parent trees, reducing NDisD. Although the authors qualify the statement with “potentially”, that does not change the fact that the argument is simplistic. The logic in the sentence about low species richness, L45-47 is also not very clear. Why should low species richness reduce distance dependence? It could potentially work in the other direction – for example, few species could support specialization on the remaining species (e.g. May 1991 argues you'd expect lower specialization by fungi on diverse host communities).

>>R: Thank you very much for your constructive suggestion. We revised these sentences and emphasized that though changes in biotic interactions may alter NDisD, but the direction of such alteration is uncertain. Lines 44-54.

L186: Were these fungal pathogens actually isolated from seeds and seedlings in this system? The references cited don't appear to mention these particular fungal pathogens (or really identify the fungal pathogens to genus). They also cover different plant species (although Liu et al 2012 considers two other *Castanopsis* species).

>>R: We agreed with your comment and deleted this sentence. Line 221.

L152: The argument for the effect of island size (and edge effects) appears to hinge on *C. sclerophylla* being edge-sensitive. Because adults are indicative of non-edge habitats, recruitment is concentrated in these areas when the forest is dominated by edges. That argument seems to indicate that we are not seeing “positive density dependence” as the authors describe it, but habitat filtering – which is a very different process and would not affect other species in the same way (e.g. an edge preferring species might end up with a pattern consistent with “negative distance dependence” because they were concentrated near the edge and away from existing conspecific adults). The outcome for biodiversity is also likely to be very different – because the habitat filtering mechanism doesn’t lead to a rare species advantage, there is no predictable impact on biodiversity without knowing about the other species in the community.

>>R: Thanks for your constructive comment. We revised discussion to point out the uncertainty of the influence of our findings to local plant diversity and acknowledged that we should expand our experiments to more plant species with diverse life history characteristics. Lines 244-253.

L263: Doesn’t follow that a preference of rodents for *C. sclerophylla* will lead to distance dependence. Yes, it makes it more likely, but distance dependence will also depend on (1) the foraging behavior of the rodents; (2) the distribution of rodent territories. There is certainly a possibility that distance dependence could be a factor here, but stating that distance dependence dominates spatial recruitment is an overreach.

Following on from my comment on L263 - It would also be helpful if the authors were more specific about the strength of the preference - “far higher” is a matter of interpretation – can you attach numbers to this preference so that they are not just in the supplement? An analysis of variance examining the proportion of the variance in predation attributed to host species might work as a means to quantify the specialization (or some measure from the host-specificity literature).

>>R: We agreed with your comment and changed “dominates” to “greatly contributes

to” (lines 310-311). In addition, we showed the attack rates (proportion of seeds being attacked by rodents) of three studied Fagaceae species and illustrated that rodents had a strong preference to attack *C. sclerophylla* seeds (lines 308-309).

L334: The formulae for the binomial models don't seem to make much sense to me – why is island size category being treated as a random effect? Given that the test is of the null hypothesis that island size has no effect on survival and the effect of distance on survival, they should be fixed effects, not random effects. It is also a little strange to have habitat type in both fixed and random intercepts. Those two terms will be almost unidentifiable (the only reason they will be partially identifiable is because of the normality constraint on the random intercepts). Additionally, what is the distinction between habitat type and experimental site size? It appears that this formulation might have been in response to my comments on the last version – however, this is not quite what I was suggesting. A formulation like

```
glmer(cbind(deaths, survs) ~ island_size * log(dtinct) + (1|year) + (1|island/experimental_site), family=binomial)
```

Might be more appropriate.

Additionally, it is the author's prerogative to decide whether to use island size as a continuous or discrete predictor. However, the former is more generalizable and does not create artificial bins with the associated idiosyncrasies (e.g. a step difference between islands of size 10 ha and 11 ha). It is not necessarily true that interactions between continuous predictors are very hard to explain. It could be argued that it is far harder to explain 3 interactions (one with each size class) than a single one. The point about non-linearities is valid, but it should be acknowledged that the authors are imposing a specific non-linear structure on their model (i.e. one that has step changes between size-classes). The consequence is that it is very difficult to generalize from the result to other contexts. For example, is a small island in their experimental system equivalent to a small island in another experiment? It is far easier to make such comparisons when size is continuous.

>>R: We really appreciate your constructive comments to improve our statistical

analyses, and we removed the random effect (1|habitat type/experimental site size) and updated the GLMMs using LN (fragment size (i.e. island size)) × LN (DTNCT) (lines 404-417), where LN represents the natural logarithm. We found a significant negative interaction between fragment size and DTNCT at each seed-seedling transition stage, showing that regression slopes between seed attack rate or seed/seedling mortality rate and DTNCT decreased with increasing fragment size (e.g. the GLMM function at seed dispersal stage was: $\text{LN}((\text{seed attack rate})/(1 - \text{seed attack rate})) = 0.13 \times \text{LN}(\text{fragment size}) + (0.02 - 0.07 \times \text{LN}(\text{fragment size})) \times \text{LN}(\text{DTNCT}) + 1.35$; see Supplementary Table 5). These results illustrated that with increasing fragment size, PDisD (when $\text{LN}(\text{fragment size}) < 0.02/0.07$) turned to NDisD (when $\text{LN}(\text{fragment size}) > 0.02/0.07$) and then the strength of NDisD increased linearly.

However, we choose to retain the results from GLMMs using fragment types (the discrete predictor), because: (1) they displayed a non-linear relationship between strength of distance-dependent mortality and fragment sizes (see Results and Fig. 2) rather than the linear trend revealed by the interaction between two continuous variables; and (2) there was a huge gap (1.1~3.9 ha for small islands; 13~51 ha for medium islands; 875 ha for the large island and > 300 ha for mainland forest patches) in fragment size among fragment types (not a continuous distribution of fragment size, lines 293-297). Since we did not describe the complete pattern of variation in strength of distance-dependent mortality, we acknowledged this limitation in Discussion and pointed out the necessity for experiments on islands with various sizes in future studies (lines 249-253).

Reviewer #3 (Remarks to the Author):

I reviewed the previous manuscript by Wang et al. As I said before, I am very impressed with the large amount of field work the authors undertook for this study, and I think the research questions are very interesting. My primary concerns were with the model structures used. While the authors have adequately addressed some of my previous concerns, I still do not think they are using the appropriate models in

some cases. I have described my concerns below. I think if the authors can address these concerns, this would be a publishable study that would be of interest to many ecologists.

The explanation of why year was kept as a random effect does not make sense: “We however still chose to set year as a random effect because: (1) our purpose was to show the change of distance-dependent mortality on small and medium islands comparing with larger habitats rather than the variation among different year; and (2) it is very difficult to explain a three-way interaction.” I was not suggesting to include any interactions with year, but simply to include it as a categorical fixed effect, rather than a random effect. It does not matter that you were not primarily interested in variation among years. The problem is that you should not include a random effect that has less than 5-6 groups/levels. In your case with only 3-4 years, it is not appropriate to treat year as a random effect. You may find this blog post useful: <https://dynamicecology.wordpress.com/2015/11/04/is-it-a-fixed-or-random-effect/>. I also recommend consulting the book by Gelman and Hill entitled “Data Analysis Using Regression and Multilevel/Hierarchical Models”.

>>R: Thank you very much for your constructive comments to improve our statistical analyses. We included year as a fixed effect in all GLMMs and LMMs (lines 383-470 in Methods), and the new results consistent with our previous conclusion (see Table 1 and Supplementary Tables 1-5).

Another issue with the current model structure is that it does not make sense to include as a random effect the term: (1|habitat type/experimental site size). I think this was done in response to one of the other reviewer’s comments, but I think the authors have misinterpreted what that reviewer was saying. They were not saying you needed to include experimental site size as a random effect. I believe they were saying that you need to include (1|year) + (1|island/site), because island size (as well as other factors) varies at the whole island level so you need to include site nested within island as a random effect, NOT that you should additionally include experimental site

size as a random effect. It does not make sense to include experimental site size as a random effect. Please remove that term (1|habitat type/experimental site size) from the models where it was included.

>>R: Thanks for pointing out a flaw in our statistical analyses. We removed the term (1|habitat type/experimental site size) from the models where it was included (lines 393-395, 448-450 and 458-460) and updated results (see Supplementary Tables 2 and 4).

I am also concerned with the analysis to examine the cause of mortality. Specifically, cause is included as a predictor variable in the model of mortality rate, but this doesn't make sense. How can mortality rate vary with cause? Were there attacked individuals that survived? If not, I don't understand this. Also, where is habitat type in these models? The goal was to test whether the mortality rate by the two causes different between habitat types. I think the model that makes sense is proportion of deaths caused by animal attack $\sim \text{LN}(\text{DTNCT}) * \text{habitat}$ and then a similar model for proportion of deaths caused by pathogen attack. For example, proportion of deaths caused by animal attack means if 100 seedlings died, and 80 of those deaths were due to animals, the proportion would be 80/100.

>>R: We updated our methods and results according to your comment (lines 418-437). The new results were still consistent with our conclusion (see Supplementary Table 3).

Other suggestions:

I think the term 'habitat type' and 'habitat size' should be changed to 'fragment size' throughout. The word habitat suggests some difference beyond size (e.g. in topography or soil nutrients).

>>R: We followed your constructive suggestion and changed "habitat type" to "fragment type" and "habitat size" to "fragment size" throughout our manuscript.

There have been some very recent papers that might be relevant to mention and cite:

Viswanathan et al. 2019 Proc Roy Soc B and Krishnadas et al. 2018 Nature Communications. Both examine effects of fragmentation on plant-pathogen interactions and the second one also looks at CNDD.

>>R: Thank you very much for the update of recent proceedings, and we cited these two publications into our manuscript. Lines 49-52, 195-197 and 244-246.

Since the Methods section comes after Results in this journal format, you need to explain the tests better here to help readers understand what you did, i.e. we examined .. and found...

>>R: Revised. We added a sentence to state the purpose and statistical analyses for each part of results. Lines 97-102, 119-121, 137-140 and 161-165.

Overall, the writing is very good. There are some small grammatical mistakes throughout the manuscript (which is completely understandable, in my opinion). I have made some suggested edits below:

Line 20 – change “strengthened” to “stronger”

>>R: Revised. Line 22.

Line 21 – change ”positive distance-dependence of mortality” to “positive distance-dependent mortality”

>>R: Revised. Line 23.

Line 31 – add “pattern of” before mortality

>>R: Added. Line 33.

Line 53 – change “frustrated” to “failed”

>>R: Revised. Line 60.

Line 54 – change “long-distance dispersed” to “dispersed long distances”

>>R: Revised. Lines 61-62.

Line 61 – Check this recent paper

>>R: Revised. We changed “no study” to “few studies”. Line 68.

Line 63- Why are the sizes for medium and small islands still listed as 10-100 ha and <10 ha, when in the response to reviewers a narrower range was given (from 13 to 51

ha for medium and from 1.1 to 3.9 ha for small islands). Please use the more accurate narrower ranges.

>>R: Revised. Lines 71-72.

Line 64 – What do you mean by “The relative and population densities”? By relative, do you mean relative abundance (ie relative to other species present)? If not, I don’t understand what ‘relative’ means here.

>>R: Revised. It is our mistake and we meant relative abundance. We revised throughout our manuscript. Lines 73-74.

Line 68 – change to ‘four consecutive year’

>>R: Revised. Line 78.

Line 71 – The first question listed could be stated better. I suggest something like: “(1) Was there evidence of negative distance dependence at early life stages in this study system?”

>>R: Revised. Lines 80-82.

Line 72 – “changed” should be ‘change’

>>R: Revised. Lines 82-83.

Line 73 – add “did” after “how”

>>R: Revised. Line 84.

Line 74 – change “responded” to “respond”

>>R: Revised. Line 84.

Line 81 (and elsewhere) – change “could hardly” to “is unlikely to”

>>R: Revised. Line 93.

Line 82 (and elsewhere) – change “survival seedlings” to “surviving seedlings”

>>R: Revised, and we changed “survival seedlings” to “surviving seedlings” across our manuscript. Lines 94-95.

Line 99 – Does “animal attack” just mean rodents? Or also insects or other species?

>>R: Animals that attacked seeds mainly included rodents but we also found some seedlings destroyed by wild boars. We therefore further explained “animal attack” in Results and Methods. Lines 118 and 365-366.

Line 111 – change “drifted” to “shifted”

>>R: Revised. Line 133.

Line 133 – change “released” to “had”

>>R: Revised. Line 165.

Line 140 – I would remove or rephrase the statement “Although the factors influencing distance-dependent plant regeneration remain unclear”. It is not clear whether you are talking about your own study here or previous studies, and it is a confusing way to start the discussion.

>>R: Removed. Lines 172-173.

Line 150 – change ‘must’ to ‘likely’

>>R: Revised. Line 183.

Line 153 – This argument suggests that overall seedling mortality should be higher on small islands than medium and large islands. Was this true?

>>R: Revised. We wanted to state that seeds dispersed far away from adult trees were likely to suffer high mortality because many of them were dispersed into edge zone. We removed this sentence because it is ambiguous and not relevant to our topic (edge effect was an important cause forming PDisD). Lines 187-188.

Line 163 – “Higher trophic levels” compared to what?

>>R: Revised. We meant “higher trophic levels compared to plants”. Line 197.

Line 196 – What does ‘less effective’ mean here? Less effective at doing what?

>>R: Revised. We meant that the influence of heterospecifics was likely to be limited because *C. sclerophylla* was the dominant tree species. Lines 231-233.

Line 203 – change “must strongly restrict” to “likely restricts”

>>R: Revised. Line 240.

Line 210 – Remove the statement “as expected by the intermediate disturbance hypothesis”. This is not correct to say that the IDH predicts this and plus the IDH is somewhat controversial, so I don’t think you should mention it at all. It is not very relevant to your findings.

>>R: Removed. Lines 253-254.

Line 239 – change ‘media’ to ‘mode’

>>R: Revised. Line 283.

Line 240 – change ‘piori’ to ‘prior’

>>R: Revised. Line 284.

Line 255 – again, I don’t understand what ‘relative density’ means? What are the units and how is this calculated?

>>R: Revised. It should be “relative abundance”. Line 301.

Line 280 – Did you test whether initial DTNCTs of placed seeds was actually consistent across all habitat types?

>>R: Yes, we tested the initial DTNCTs among fragment types and found no difference using LMMs, and the results were indicated in Methods. Lines 330-332 and Supplementary Table 1.

Line 283 – You should write out numbers when it is the first word at the start of a sentence, so “Fifty” instead of “50”

>>R: Revised. Line 333.

Line 284 – change “may hardly occur” to “is unlikely to occur”

>>R: Revised. Line 335.

Line 300 – change “could hardly” to “are unlikely to”

>>R: Revised. Lines 350-351.

Line 378-379 – I think by ‘rodent density’ you mean the proportion of traps with rodents, but rodent density per unit area. Please clarify because it does not make sense to use binomial errors unless your dependent variable is a proportion.

>>R: Revised. We used the proportion of cages that captured rodents per night as the estimation of rodent density because no rodents were recaptured during our four-year experiments. Lines 378-381 and 454-455.

REVIEWERS' COMMENTS:

Reviewer #1 (Remarks to the Author):

I would like to start off by thanking the authors for altering their modeling approach as suggested by me and the other reviewer, and also for considering treating fragment area as a continuous term. I think the statistical analyses are now adequate.

I have a few additional small comments that I would like to make on the paper.

L32: The authors refer to distance dependence as negative, but it is probably worth noting that this terminology could end up being confusing. The relationship between distance and *mortality* is expected to be negative according to the Janzen-Connell hypothesis. However, the literature is largely couched in terms of the effects of neighborhood on *survival* (e.g. CNDD = conspecific negative density dependence of survival). A reader might get confused that the authors are suggesting that survival rather than mortality has a negative relationship with distance, so perhaps replacing NDisD with NDisDM would help?

L47: I am not sure the papers referred to back up this statement (Ewers & Didham, 2006; Krishnadas et al, 2018, Mendes et al, 2015) are the best papers to show that specialized consumers are lost from fragments. Some better papers might be Gravel et al, Ecol. Letts, 2013 and Bagchi et al. 2018. Oecologia, which both address this topic explicitly (rather than patterns that could be explained by lost specialists).

L55: Many plant species will benefit from being in an edge environment. The current statement is too general.

L58: The suggestion that edges provide a generally hostile environment for seed and seedling survival is not accurate – many species preferentially recruit near edges.

L68: “was high”

L187: What is the basis of the assumption that rodents would be mostly near adult *C. sclerophylla* trees?

L195: why would large and medium sized islands be better at supporting pathogens? Is there a microclimate effect? Viswanathan et al (2019, Biology Letters) found a pattern of reduced pathogen attack in small forest fragments, potentially due to altered microclimates (they couldn't clearly link their result to altered microclimate however). There may be other papers that show these patterns too – I would like to see some more support for this suggestion.

Reviewer #3 (Remarks to the Author):

The manuscript is much improved, and the authors have dealt with all of my previous

comments. This is a nice study that obviously took a lot of hard work!

I found a few very small grammar/wording issues that should be corrected in the final version:

-Line 68: Change "were" to "was"

-Lines 110-112: in the sentence "where there was significant PDisD and NDisD in proportions of dead seeds and seedlings caused by pathogen infestation and animal attack, respectively", it is not clear if respectively is referring to PDisD and NDisD and/or to seeds and seedlings. I suggest making this a separate sentence and changing the wording to make it clear.

-Line 211 - "thresholds" should be "threshold"

-Line 214 - remove "local dominant" and change to "plant species"

-Line 217 - remove "huge"

-Line 556 - change "helps" to "help"

Responses to comments

Reviewers' comments:

Reviewer #1 (Remarks to the Author):

I would like to start off by thanking the authors for altering their modeling approach as suggested by me and the other reviewer, and also for considering treating fragment area as a continuous term. I think the statistical analyses are now adequate. I have a few additional small comments that I would like to make on the paper.

L32: The authors refer to distance dependence as negative, but it is probably worth noting that this terminology could end up being confusing. The relationship between distance and *mortality* is expected to be negative according to the Janzen-Connell hypothesis. However, the literature is largely couched in terms of the effects of neighborhood on *survival* (e.g. CNDD = conspecific negative density dependence of survival). A reader might get confused that the authors are suggesting that survival rather than mortality has a negative relationship with distance, so perhaps replacing NDisD with NDisDM would help?

>>R: Thank you very much for this constructive suggestion, and we have replaced NDisD with NDisDM and PDisD with PDisDM throughout the manuscript.

L47: I am not sure the papers referred to back up this statement (Ewers & Didham, 2006; Krishnadas et al, 2018, Mendes et al, 2015) are the best papers to show that specialized consumers are lost from fragments. Some better papers might be Gravel et al, Ecol. Letts, 2013 and Bagchi et al. 2018. Oecologia, which both address this topic explicitly (rather than patterns that could be explained by lost specialists).

>>R: We have cited these two papers to further ~~support~~ address our ~~statement~~ topic.
Line 50.

L55: Many plant species will benefit from being in an edge environment. The current

statement is too general.

>>R: Revised. Lines 54-58.

L58: The suggestion that edges provide a generally hostile environment for seed and seedling survival is not accurate – many species preferentially recruit near edges.

>>R: Revised. In the revised manuscript, we focused on the plants adapting to the interior environments in forests (Lines 58-62).

L68: “was high”

>>R: Revised (Line 71).

L187: What is the basis of the assumption that rodents would be mostly near adult *C. sclerophylla* trees?

>>R: Revised. We added some information to support the assumption (Lines 196-201).

L195: Why would large and medium sized islands be better at supporting pathogens? Is there a microclimate effect? Viswanathan et al (2019, Biology Letters) found a pattern of reduced pathogen attack in small forest fragments, potentially due to altered microclimates (they couldn't clearly link their result to altered microclimate however). There may be other papers that show these patterns too – I would like to see some more support for this suggestion.

>>R: Yes, there were more cases supporting the pattern. In the revised manuscript, we added two case studies, one suggesting that pathogenic fungi caused death decreased from the edge to forest core (Ichihara & Yamaji 2009), and the other suggesting that soil fungus biomass increased from the edge to forest core (Kageyama et al. 2008). (Lines 205-209).

Reviewer #3 (Remarks to the Author):

The manuscript is much improved, and the authors have dealt with all of my previous

comments. This is a nice study that obviously took a lot of hard work! I found a few very small grammar/wording issues that should be corrected in the final version:

Line 68: Change "were" to "was"

>>R: Revised (Line 71).

Lines 110-112: in the sentence "where there was significant PDisD and NDisD in proportions of dead seeds and seedlings caused by pathogen infestation and animal attack, respectively", it is not clear if respectively is referring to PDisD and NDisD and/or to seeds and seedlings. I suggest making this a separate sentence and changing the wording to make it clear.

>>R: Revised (Lines 119-124).

Line 211 - "thresholds" should be "threshold"

>>R: Revised (Line 222).

Line 214 - remove "local dominant" and change to "plant species"

>>R: Revised (Line 226).

Line 217 - remove "huge"

>>R: Removed (Line 228).

Line 556 - change "helps" to "help"

>>R: Revised (Line 579).